# JCPyV VP1 Mutations in Progressive Multifocal Leukoencephalopathy: Altering Tropism or Mediating Immune Evasion?

**DOI:** 10.3390/v12101156

**Published:** 2020-10-12

**Authors:** Matthew D. Lauver, Aron E. Lukacher

**Affiliations:** Department of Microbiology and Immunology, Penn State College of Medicine, Hershey, PA 17033, USA; mvl5637@psu.edu

**Keywords:** polyomavirus, JCPyV, MuPyV, VP1, antibody escape, persistent infection, Progressive multifocal leukoencephalopathy (PML)

## Abstract

Polyomaviruses are ubiquitous human pathogens that cause lifelong, asymptomatic infections in healthy individuals. Although these viruses are restrained by an intact immune system, immunocompromised individuals are at risk for developing severe diseases driven by resurgent viral replication. In particular, loss of immune control over JC polyomavirus can lead to the development of the demyelinating brain disease progressive multifocal leukoencephalopathy (PML). Viral isolates from PML patients frequently carry point mutations in the major capsid protein, VP1, which mediates virion binding to cellular glycan receptors. Because polyomaviruses are non-enveloped, VP1 is also the target of the host’s neutralizing antibody response. Thus, VP1 mutations could affect tropism and/or recognition by polyomavirus-specific antibodies. How these mutations predispose susceptible individuals to PML and other JCPyV-associated CNS diseases remains to be fully elucidated. Here, we review the current understanding of polyomavirus capsid mutations and their effects on viral tropism, immune evasion, and virulence.

## 1. Introduction

Polyomaviruses (PyVs) are ubiquitous members of both human and non-human viromes. First discovered in mice as an infectious oncogenic agent, polyomaviruses have since been identified in a wide variety of mammals and birds, and more recently in fish and arthropods [1,2,3]. These viruses have coevolved with their animal hosts and typically cause a lifelong asymptomatic infection with persistent viral shedding [1]. Fourteen human polyomaviruses have been identified to date, with estimated seroprevalence rates ranging from 4–98% in the general population and seropositivity rates increasing with age [4]. The route of transmission is currently not known, but is likely urine/fecal-oral as polyomaviruses are detected in sewage and are highly stable under these conditions [5,6]. There is also evidence that primary infection by PyVs may occur via respiratory inhalation [7]. Although the majority of polyomavirus infections are asymptomatic, several polyomaviruses cause severe diseases in immunocompromised individuals. BKPyV is responsible for BKPyV-associated nephropathy (BKVN), a major cause of dysfunction and loss of kidney allografts, and hemorrhagic cystitis in bone marrow transplant recipients [8,9,10,11]. MCPyV is the etiologic agent for Merkel cell carcinoma, an aggressive cutaneous malignancy [12]. JCPyV causes the oft-fatal demyelinating brain disease Progressive Multifocal Leukoencephalopathy (PML) [13]. JCPyV seroprevalance is >60%; in healthy individuals, JCPyV persists predominantly in the urinary tract, with intermittent virus shedding into the urine during pregnancy [4,14,15,16]. The state of persistent polyomavirus infection (i.e., low-level “smoldering” vs. latency-reactivation) and the viral and immunologic determinants that prevent resurgence are unknown. This review focuses on our current understanding of JCPyV capsid protein mutations in PML and how they relate to disease development.

## 2. The Polyomavirus Lifecycle

PyVs are non-enveloped viruses with circular, double-stranded DNA genomes of approximately 5000 bp. The viral genome, packaged within the capsid as a supercoiled minichromosome, is divided into two coding regions: (1) the early region contains genes for the 2–3 major non-structural T proteins (small T antigen (ST), middle T antigen (MT), and large T antigen (LT)); and (2) the late gene region encodes the VP1-3 capsid proteins and for SV40-clade polyomaviruses (e.g., SV40, BKPyV, and JCPyV) the non-structural agnoprotein. The early and genes are transcribed from a single early and late promoter, respectively, with their open reading frames joined by a shared polyA region. The early and late transcription start sites are separated by the non-coding control region (NCCR), an approximately 500 bp segment containing the origin of replication and binding sites for host cell transcription factors that regulate viral replication in cells of different lineages.

The polyomavirus capsid consists of 360 copies of VP1 arranged into 72 pentamers called “capsomers” in a T = 7 d icosahedral configuration. The internal face of each capsomer associates with a single copy of a minor capsid protein, VP2 or VP3. The secondary structure of VP1 is organized as antiparallel β strands (BIDG and CHEF) which form a structure commonly referred to as a jelly roll [17]. The loops connecting several of these strands (BC, DE, EF, and HI) form the majority of the external surface of the capsid (Figure 1). This surface forms a depression which is responsible for receptor binding. Polyomaviruses bind to sialyated glycans, with strains within each polyomavirus family varying in recognition of sialic acids having different linkages to glycolipid and/or glycoprotein backbones. The canonical receptor for JCPyV is the sialylated oligosaccharide lactoseries tetrasaccharide c (LSTc), which interacts with residues in the BC and HI loops of VP1 [18,19]. VP1 mutations disrupting the interactions with LSTc have profound effects on viral infectivity [18]. After receptor binding, JCPyV entry is facilitated by the 5-HT_2_ serotonin receptor [20,21]. 

After receptor binding and internalization, virus particles traffic in single virion encapsulated vesicles to the endoplasmic reticulum (ER) [23,24]. Capsid conformational changes and interactions with ER and cytosolic host proteins result in destabilization of VP1 capsomers and ejection of virions across the ER membrane into the cytosol [25,26,27,28]. The virus capsid disassembles and the viral genome is imported into the nucleus to begin early gene expression [29,30]. Production of the T antigens occurs from alternative splicing of a single RNA transcript. PyVs typically encode large and small T antigens; rodent PyVs also encode the middle T antigen, an oncoprotein likened to a constitutively active tyrosine growth factor receptor [31]. Large T antigen directly binds to and sequesters the cellular retinoblastoma proteins and, depending on the PyV family, binds p53. PyV LT-mediated inactivation of these tumor suppressor proteins engages cell cycling, allowing the virus to highjack the host cell DNA replication apparatus to synthesize its genome. Genome replication leads to transcription of the late genes, which form the capsid to contain the nascent viral genomes and complete the virus replication cycle. Some polyomaviruses, including JCPyV and BKPyV, encode an additional late gene, the agnoprotein; recent work suggests that that agnoprotein interdicts the cell’s innate immune antiviral defenses [32]. Release of progeny PyV virions was long considered to be coincident with host cell lysis. Recent work from the Atwood group has raised the possibility that infectious virions ensconced within vesicles are released from host cells prior to cell death, at least by JCPyV in glial cells and choroid plexus epithelial cells [33,34].

Although PyVs cause lifelong infections, they do not encode genes dedicated to sequestering the virus in a state of latency. All PyVs examined to date encode microRNAs complementary to the LT mRNA transcripts [35,36]. LT binds to NCCR sites as hexameric complexes, where it engages its helicase activity to drive viral genomic DNA synthesis. By eliminating LT transcripts, microRNAs act to reduce viral genome replication and concomitantly remove/reduce epitope targets for virus-specific T-cells [37,38,39]. Alternatively, the concept of “proteostatic viral latency” has been advanced by Kwun, Moore, and coworkers whereby particular host cell E3 ubiquitin ligases instigate proteasome-mediated degradation of LT in MCPyV-infected cells [40,41,42]. A common theme for both mechanisms is to keep copy numbers of LT low. Whether the resultant dampened viral replication is sufficient to hold the virus in a true or partial state of latency in vivo, and whether different PyVs use alternate mechanisms to maintain latency, are unclear.

A non-mutually exclusive possibility is that PyV may persist as a smoldering infection; i.e., continuous low levels of viral replication sufficient to allow virus spread to maintain persistence within a host and transmission to new hosts, while avoiding tissue injury. It is also tempting to speculate that PyV virions may continually spread cell-to-cell via extracellular vesicles (EVs), perhaps without killing the infected host cell. This low level, silent infectious state implies a delicate balance between a continually replicating virus and host antiviral immunity. PyV infections are highly immunogenic, inducing potent cellular and humoral immune responses [43,44,45,46]. Virus-specific T-cells and neutralizing antibodies are critical for initially controlling PyV infection and maintaining immune control over the virus. The absence of T-cells or antibody results in severe PyV infection, and the loss of adaptive immunity during persistent infection results in viral resurgence [47,48,49,50,51]. Mirroring this, immunosuppressive conditions can set the stage for a loss of immune control over JCPyV and incur severe morbidity and mortality.

## 3. PML: A Consequence of Immune Perturbation

Originally identified as a rare complication of hematological malignancies, PML’s incidence increased dramatically with the emergence of the human immunodeficiency virus and acquired immune deficiency syndrome (HIV/AIDS) epidemic, such that up to 5% of AIDS patients developed PML in the pre-antiretroviral era [52,53,54]. PML as an AIDS-defining disease vanished with the advent of antiretrovirals, but administration of specific immunomodulators for treatment of autoimmune and inflammatory diseases have led to an uptick in PML cases. In particular, patients treated with natalizumab (targeting α4 subunit integrins) for multiple sclerosis, rituximab (targeting CD20) for non-Hodgkin’s lymphoma and chronic lymphocytic leukemia, or efalizumab (targeting LFA-1) for plaque psoriasis put patients at increased risk for PML [55,56,57,58,59,60].

PML results from resurgent JCPyV replication in the central nervous system (CNS), likely within astrocytes, leading to viral spread to and death of nearby myelin-producing oligodendrocytes. Infected astrocytes display an atypical or “bizarre” morphology, showing enlarged and misshapen nuclei, consistent with a transformed phenotype; evidence suggests that astrocytes support lytic JCPyV infection and suffer necrotic cell death [61,62,63,64,65]. In contrast, JCPyV replication in oligodendrocytes may be non-productive. Whether loss of oligodendrocytes results from apoptosis caused by forced cell cycle entry by a terminally differentiated cell or reflects activation of a non-apoptotic cell death pathway remains to be determined [61,66,67,68]. Oligodendrocyte loss causes axon demyelination and the foci of damage seen by MRI as hyperintense lesions on T2-weighted and fluid-attenuated inversion recovery (FLAIR) images [69,70,71]. Because no anti-PyV agents are available, the current mainstay treatment for PML involves cessation of immunosuppression to allow immunologic control of the infection. However, doing so is often complicated by a high-mortality immune reconstitution inflammatory syndrome (IRIS) due to a robust specific and bystander immune responses against JCPyV CNS infection [72,73,74,75]. The mechanisms responsible for viral outgrowth, dissemination, and CNS infection are largely unknown. This black box regarding key aspects of PML development and pathogenesis lays down a roadblock to stratifying patients for PML risk, identifying conditions/therapeutics that increase PML risk, and developing effective treatments for PML.

A hallmark of PML pathogenesis is the emergence of the viral variants that are unique from the circulating (called wild type or archetype) JCPyV strains [76,77,78,79]. These PML-associated JCPyV (JCPyV-PML) variants contain alterations to the NCCR in the form of inversions, duplications, translocations, and deletions [77,78,79]. JCPyVs with such “rearranged” NCCRs are inextricably tied to PML development and enhance viral replication and infection in glial cells by modifying transcription factor binding in the NCCR [80,81,82]. Additionally, JCPyV-PML variants frequently have discrete, non-synonymous point mutations in the solvent-exposed loops of the VP1 capsid protein [76,83,84,85]. These mutations are only detected in isolates from PML patients [86]. This strong association between these VP1 mutations and PML indicates that they could participate in PML pathogenesis. Understanding the functions and effects of these mutations in the setting of PyV infection can illuminate portions of the PML black box.

## 4. Of Men and Mice: Models of PyV Infection

PyVs are highly species-specific, such that viral replication is limited to their natural host reservoir. This tight host range results from specific recognition of the viral origin of replication by the host DNA polα-primase complex [87,88]. SV40, a non-human primate PyV, has been extensively used in vitro for studying the PyV lifecycle [89]. SV40 can cause a PML-like CNS disease in immunosuppressed macaques, but prohibitive costs and the lack of inbred and transgenic models limit its applications [48,90,91,92]. Due to its ease of propagation in tissue culture and in the mouse, the best immunologically and genetically characterized mammalian model, mouse polyomavirus (MuPyV) is the main tractable system for studying PyV pathogenesis. MuPyV has provided numerous insights into viral carcinogenesis and the immune response to PyV infection [93,94,95]. In addition, studies of VP1 mutations in MuPyV have revealed profound effects of these mutations on viral tropism and pathogenesis [96,97,98].

A human glia chimeric mouse model for JCPyV infection has recently been described [67]. These mice contain human glia as the result of engrafting human glial progenitor cells (GPCs) into neonatal hypomyelinated mice lacking an adaptive immune system (*Mbps^shi/shi^Rag2*^−/−^). The engrafted glia populate and myelinate axons in the brain, giving rise to an adult mouse with human glia. These mice support JCPyV infection and develop demyelination; however, infection is confined to the engrafted human glia. These mice lack T- and B-cells, obviating investigation of adaptive anti-PyV immunity [67].

## 5. VP1 Mutations: Altering Viral Tropism

JCPyV-PML mutations are predominantly located in the external loops of VP1 and overlap many of the reported LSTc-interacting residues (Figure 2). In particular, three of the most commonly mutated residues (L54, S266, and S268; each frequently mutated to phenylalanine in JCPyV-PML) participate in contacting the sialyated moiety of LSTc during receptor binding. Substitutions at these residues that disrupt these interactions (including the mutations seen in PML) largely abrogate the ability of VP1 to bind LSTc [18,99]. In turn, these JCPyV-PML mutations render the virus incapable of infecting immortalized human glial cells and prevent sialic acid-mediated hemagglutination [84,99]. Instead, these mutations restrict virus binding to non-sialylated glycosaminoglycans, which allows these VP1 mutated viruses to retain infectivity in some cell lines [100]. However, the effect(s) of these mutations on viral tropism, particularly in vivo, is undetermined. Viruses bearing the L54F, K59E, and S268F mutations display WT levels of infectivity in primary human astrocytes, oligodendrocytes, and GPCs. Furthermore, in *Mbps^shi/shi^Rag2^−/−^* mice engrafted with human GPCs, L54F and S268F mutant viruses generate robust infection and demyelination similar to WT virus [67]. These data suggest that although these VP1 mutations may impair infection of some glial cells in tissue culture, they do not significantly handicap the ability of the virus to infect glia in vivo.

Atwood and coworkers have recently described a plausible mechanism that resolves this discrepancy between infection by VP1 mutant viruses in vivo but lack of infection in vitro. JCPyV was found to be capable of spreading cell-to-cell via EVs released from infected cells [33,34]. Both WT and VP1 mutant JCPyV can be released in vesicles and infect cells, and infection is independent of the presence of the LSTc and 5-HT_2_ receptors. Moreover, envelopment of virions in EVs shields them from neutralizing VP1-specific antibodies. Immortalized glia and primary choroid plexus epithelial cells can produce virus-containing EVs, which can infect other glia. Receptor-independent infection of glia provides a mechanism by which VP1 mutant viruses could infect glial cells despite defects in receptor binding that negate direct infection by free virions. How EVs bind, undergo internalization, and release their encapsulated PyV virions to enter the infectious pathway remain to be elucidated.

Less is known about the effects of these JCPyV-PML VP1 mutations on kidney infection. Several of the mutations disrupt binding to kidney tubular epithelial cells [84]. Viral isolates from the urine of PML patients predominantly have archetype VP1 sequences, although mutant VP1 sequences are detected in the patients’ blood and CSF [84,85]. This difference in regionalization suggests that these mutations and alterations in receptor binding disadvantage virus growth in the kidney. As a result, the parental WT virus remains the dominant species in the kidney despite the viremia and brain disease induced by the mutant virus. This hit to viral fitness in the kidney by JCPyV-PML VP1 mutations is further supported by reports of JCPyV-driven nephropathy, in which viral urine isolates bear a wild type VP1 sequence or mutations distinct from those seen in JCPyV-PML isolates [102]. These data suggest the mutations impair viral infection in the kidney, but retain the ability to cause brain infection and pathology.

Two well-characterized PyV VP1 mutations in MuPyV known to affect viral tropism are the E91G and V296A mutations located in the BC and HI loops, respectively. E91 and V296 both participate in receptor binding, but these two mutations have drastically different effects on viral pathogenesis. MuPyVs carrying E91G exhibit severely impaired kidney infection and the profile of tumors they induce shifts from those of epithelial to mesenchymal lineage [96,103]. This impairment likely results from increased affinity by these E91G VP1 mutant MuPyVs for branched-chain sialyloligosaccharides, which act as pseudoreceptors [97]. This possibility is supported by evidence that E91G mutant viruses bind cell surface glycoproteins, which divert the virus away from glycolipid receptors and entry into the productive infection pathway [104]. MuPyVs carrying the VP1 mutation V296A, conversely, exceed WT virus in replicative efficiency in the kidney and kill newborn-inoculated mice as neonates [98]. This virulence is due to decreased affinity for sialylated receptors, resulting in increased viral spread [97]. VP1 sequences in MuPyV isolates from feral mice are invariably E91 and V296, which fits with the idea that such VP1 sequences enable efficient inter-mouse transmission of the virus [105]. Notably, V296 of MuPyV VP1 corresponds to S268 of JCPyV-PML VP1 [86]. MuPyVs carrying a V296F VP1 mutation infect similar glial cell types and replicate to equivalent levels as parental virus in the brain. However, this mutation impairs viral infection and persistence in the kidney, resulting in the absence of severe kidney infection in immunodeficient mice and poor shedding in the urine [106]. Disruption of kidney infection and retained glia infection suggests that the emergence of JCPyV-PML VP1 variants is driven by a factor(s) other than enhanced viral tropism or replication.

## 6. VP1 Mutations: Facilitating Immune Evasion

Evasion of host neutralizing antibody (nAb) responses applies a powerful selective pressure during persistent viral infection, but may incur a hit to viral replicative fitness and alter host cell tropism. Examples include antibody-escape mutants appearing over the course of HIV and hepatitis C virus (HCV) infection and during hepatitis B virus (HBV) reactivation [107,108,109,110]. Antibody escape mediated by otherwise deleterious mutations can promote the emergence of the virus variants with impaired infectivity. Despite this impairment, these mutant viruses can outcompete the parental virus because of their resistance to the host’s nAb response. Examples of this selection are seen in a variety of persistent infections. HIV mutants arise during the course of infection that evade host nABs at the cost of decreased replication [111]. nAb escape in HBV is mediated by mutations that concomitantly impair virion production [110,112,113]. Selection of nAb resistance over the course of HCV infection promotes the outgrowth of escape mutants with progressively impaired receptor binding, even to the point of eliminating viral persistence [114]. It is therefore plausible that antibody escape could promote the outgrowth of JCPyV VP1 mutants at the cost of tropism for the urinary tract.

In kidney transplant patients, poor immune control over BKPyV can lead to BKVN and transplant loss [8,9,10]. Although most individuals are seropositive for the BKPyV serotype I, many individuals lack antibodies recognizing the other BKPyV serotypes [115,116]. Transplant of a kidney infected with a BKPyV serotype different from that of the recipient is suggested to initiate a loss of immune control leading to BKVN [117,118]. The majority of VP1 variation between serotypes occurs in the BC loop, in particular residues 61–82 [116]. Many of these residues are also sites of mutations in viral isolates from kidney transplant patients with increased viremia or BKVN. A subset of these mutations are conversions of the residue to that of a different serotype, for instance D61N, E72K, D76N, D/E81D/E/Q, or H/N138H/N [119,120,121,122]. These inter-serotype mutations suggest a potential evasion of the patients’ VP1 nAb response by partial conversion of the virus to a different serotype. Other mutations are not present in any of the major serotypes, including E72Q and D76E/H. It is likely that these achieve the same end result of nAb evasion by disrupting dominant epitopes in the BC loop, and some hinder neutralization by patient-matched sera [123]. Evidence suggests that these nAb evasion mutations come at a cost, however. These mutations alter receptor usage and engagement of sialyated receptors, as seen by alterations in viral hemagglutination of red blood cells and impaired infectivity in some cell lines [121,123].

The region(s) of JCPyV VP1 targeted by the host’s nAb response has yet to be elucidated. Our current understanding of the neutralizing epitopes in VP1 comes from studying VP1-specific monoclonal antibodies (mAbs). Several JCPyV or SV40 VP1 mAbs recognize residues in the BC loop adjacent to the receptor binding pocket, as well as residues in the DE and EF loops [124,125]. A recently reported JCPyV-BKPyV cross-neutralizing antibody binds at the three-fold axis of symmetry across VP1 molecules from three separate capsomers. The VP1 residues engaged by the antibody are highly conserved between JCPyV and BKPyV, accounting for this antibody’s ability to neutralize both viruses. The majority of this antibody’s epitope lies in the EF loop, distant from the receptor binding pocket and sites of PML mutations. However, a portion of the epitope contains several residues in the BC loop in the immediate vicinity of the frequently mutated L54 residue [126]. JCPyV mAbs generated from a natalizumab-treated PML patient provide evidence for a dominant nAb target in VP1 [127]. These mAbs all block hemagglutination by JCPyV, suggesting that they bind in or around the sialic acid biding pocket and BC/HI loops [100]. A neutralizing mAb against MuPyV VP1 binds the BC and HI loops of VP1 and neutralizes the virus by occluding the receptor binding pocket and engaging VP1 residues crucial for sialic acid binding. Interestingly, this VP1 mAb blocks the binding of >75% of the endogenous VP1 antibody response in mice to MuPyV [106]. Thus, its epitope in the BC and HI loops may represent a dominant target of the PyV antibody response in mice. The epitopes of these mAbs suggest the VP1 loops surrounding the receptor binding pocket are a common target of PyV nAbs.

Two studies provide evidence that JCPyV-PML VP1 mutations disrupt recognition by host nAbs. One study examined the neutralization of pseudoviruses displaying the PML VP1 mutations L45F, S266F, or S268F by sera from healthy individuals and PML patients [101]. In the absence of immune suppression, over 15% of healthy donors’ sera failed to neutralize one or more of these mutant pseudoviruses. In longitudinal testing of neutralization by sera from PML patients, all patient sera failed to neutralize the cognate mutant pseudovirus prior to the onset of PML, despite being able to neutralize WT VP1 pseudovirus. Patients who had progressive and ultimately fatal disease never developed neutralizing sera, but patients who survived developed neutralizing sera to their mutant VP1 over time [101]. In a concurrent study, sera from healthy donors, natalizumab-treated patients without PML, with PML, or with PML-IRIS were tested for their ability to recognize WT and mutant VP1 pentamers [127]. Serum recognition of L54F and S268F pentamers was reduced in PML patients, and sera from all patient groups had impaired binding to S266F. Multiple mAbs generated from a PML-IRIS patient showed deficiencies in binding L54F, S266F, or S268F VP1. Further evidence of nAb evasion by VP1 mutations is seen in the MuPyV V296F mutation in the HI loop of VP1 [106]. This mutation confers resistance a VP1 mAb, which neutralizes the parental virus by binding the BC and HI loops. The V296F mutation confers resistance by blocking binding of this mAb to VP1, through steric collision of the phenylalanine side chain with the heavy chain of the mAb. Several other MuPyV VP1 mutations that correspond to JCPyV-PML mutations also disrupt binding by this mAb. Together, these studies support the hypothesis that these mutations disrupt dominant nAb targets. Individuals whose nAb responses fail to recognize these VP1 mutant JCPyVs prior to immune suppression appear to be at risk for the emergence of JCPyV mutants and development of PML.

## 7. How Do VP1 Mutations Arise?

PyVs have long been thought to have a low genetic mutation rate, because they commandeer the host’s high-fidelity DNA replication machinery to replicate their genomes. The presence of VP1 mutations in BKPyV and JCPyV isolates from BKVN and PML patients, as well as rearranged NCCRs in JCPyV-PML, supports the idea that some level of mutagenesis of PyV genomic DNA is inherent in its replication process. In fact, BKPyV exists as a quasispecies even in healthy individuals. JCPyV quasispecies are also found in PML patients, raising the likelihood that a mixture of JCPyV variants emerge in individuals without PML [128,129,130]. The tissue, cell type(s), and mechanisms responsible for generating VP1 mutations are not known; for JCPyV, kidney epithelia and certain glia have been suggested as possible locations [67,84]. PyV antibody-escape mutants can arise by serial passage in epithelial cells, indicating that these cells are capable of generating VP1 mutations [106,126]. Human glia engrafted in *Mbps^shi/shi^Rag2*^−/−^ mice also generate VP1 point mutations in JCPyV [67]. Apoliproprotein B Editing Complex (APOBEC)3 cytidine deaminases have been implicated in the generation of VP1 mutations in BKVN [121]. APOBEC proteins or similar cytidine deaminases engaged in antiviral defense may unwittingly generate antibody-escape mutations in JCPyV VP1. It is likely that the mutagenic cells reside outside the CNS, such that VP1 mutant viruses would undergo selection by circulating VP1 antibodies excluded by the blood–brain barrier (BBB). Yet, JCPyV DNA has been detected in brains of individuals dying of non-PML diseases, leaving open the possibility that inflammatory conditions impacting BBB integrity may allow access of antiviral antibodies and selection of escape-variants generated in the CNS [14,131,132].

## 8. Immune Suppression Setting the Stage for Antibody Escape

Although evidence supports JCPyV-PML VP1 mutations as facilitating antibody escape, conditions predisposing patients to PML (e.g., idiopathic CD4 lymphopenia, HI/AIDS, natalizumab) predominantly affect the patient’s CD4 T-cell compartment. One explanation for the connection between T-cell suppression and the emergence of antibody escape VP1 mutations is provided by work from Ray et al. [101]. Some healthy individuals have “blind spots” for neutralizing certain VP1 mutants, and patients who develop VP1 mutations and PML have this blind spot for their cognate mutation prior to PML diagnosis. These findings suggest that PML patients start off with a nAb response inherently vulnerable to escape mutations. In healthy individuals, an intact immune system can compensate for this weakness; e.g., nAb escape mutations offer no significant advantage in the face of a robust anti-PyV T-cell response. However, weakened T-cell-mediated protection, whether by HIV/AIDS or immunomodulatory therapies, could shift the burden of viral control largely to the nAb response. A pre-existing hole in the epitope repertoire of the anti-JCPyV nAb response could then render an individual vulnerable to the outgrowth of an escape mutant, as the adaptive immune response has little recompense for dealing with the emerging variant. This has been seen experimentally in lymphocytic choriomeningitis virus infection in mice, where CD8 T-cell deficiency during infection allows for the emergence of nAb escape mutants [133]. Likewise, immune suppression can promote HBV reactivation mediated by nAb escape mutants [110].

## 9. Why Don’t Patients Make a New nAb Response to Mutant VP1s?

The emergence and outgrowth of a JCPyV antibody-escape variant should promote the development of a nAb response to the mutant, as has been reported in HCV infection [114]. PML patients have increased antibody levels toward archetype VP1, which can be detected before the development and diagnosis of PML [127,134,135]. This increase is seen despite evidence showing that PML patient sera are unable to recognize and neutralize the nascent VP1 mutant virus [101,127]. One explanation for this is that rising levels of VP1 mutant virus stimulate existing antibody-producing cells that strongly recognize the parental virus but weakly recognize the mutant. These cells proceed to produce VP1-specific antibodies that raise the patient’s JCPyV-specific antibody titers, but do little to temper the accelerating infection driven by the mutant virus. This possibility could explain why high JCPyV-specific antibody titers in PML patients seemingly show no benefit [134,136]. Indeed, bolstered immunity to a previously seen pathogen when encountering a related pathogen, called “original antigenic sin,” is a well-recognized property of antibody responses to influenza virus infections [137,138,139].

Can antibodies to VP1 mutated epitopes be coaxed in patients with PML? Three patients that received JCPyV VLP immunization together with interleukin-7 and imiquimod [a toll-like receptor (TLR)7 agonist] showed improvement in PML outcome [101,140]. For one patient, this improvement was accompanied by a marked improvement in neutralizing titers against the patient’s cognate mutant JCPyV; this was not examined in the other two patients. While it cannot be determined whether this increase in titer was responsible for the patients’ improvement, it is notable that this treatment was able to drive the production of mutant-specific nAbs [101]. This indicates that under the proper conditions, PML patients can overcome their serologic blindness to their VP1 mutant JCPyV. VLP immunization of at-risk patients could improve coverage of these nAb epitope blind spots and remove the opportunity for mutant virus outgrowth.

It remains unclear what factors are necessary for the generation of these VP1 mutant-specific nAbs. It seems unlikely that exposure solely to the VP1 immunogen is the essential antecedent event, as these PML patients had elevated JCPyV burden and theoretically an abundance of VP1 antigen. Instead, the imiquimod adjuvant could provide the necessary stimulus to overcome this block in mutant-recognizing antibody production. Notably, all three patients had an underlying CD4 T-cell lymphopenia. Experimental evidence implicates TLR co-signaling with B-cell receptors as a necessary step to elicit PyV antibody responses during CD4 T-cell insufficiency. Using MuPyV, Szomolanyi-Tsuda and coworkers showed that PyVs are capable of generating protective CD4 T-cell-independent (TI) IgG responses dependent on signaling through the adapter molecule MyD88 [45,46]. MyD88 is necessary for signal transduction by most TLRs, including TLR7 [141]. Imiquimod treatment may boost the patients’ VP1 TI antibody response and drive the de novo development of VP1 mutant-neutralizing antibodies. It is unclear whether these nAbs recognize VP1 at the site of the mutant residue or target a separate, unmutated region of the virus capsid that is conserved between the parental and PML mutant JCPyVs. Further investigation into factors that control PyV humoral immunity and defining VP1 antibody epitopes are needed to understand the mechanism of action for these therapies.

## 10. Early Detection: Can VP1 Mutations and Antibody Escape Indicate At-Risk Patients?

The risk factors for PML remain poorly defined. JCPyV seropositivity, history of immunosuppression, and >24 months of treatment are the known risk factors for PML in natalizumab-treated patients, but <3% of patients who fall into this high-risk category develop PML [142,143,144,145]. Likewise, PML occurred at a frequency of <5% in HIV/AIDS patients [53]. This low incidence indicates that other yet-to-be-defined criteria are needed to identify an individual’s risk for developing PML. Given the inability of sera from some healthy individuals to recognize certain VP1 mutant viruses and similar inability of PML patient sera to neutralize cognate mutant virus, a serologic blind spot for certain VP1 mutations is a likely high-risk factor for PML [101,127]. Serologic examination for patients with these VP1 antibody epitope blind spots prior to immunomodulatory treatment or who develop them during the course of treatment could better stratify patients for PML risk.

With viremia as a likely prerequisite for CNS infection and PML, longitudinal screening of blood for VP1 mutant JCPyV may identify those patients at highest risk for PML. Sensitive next-generation sequencing of peripheral blood for VP1 mutations could inform interventions well before clinical and brain-imaging signs of PML manifest [146]. Although cessation of immunosuppression often leads to PML-IRIS due to extensive JCPyV brain infection, early detection of VP1 mutations could circumvent this complication by reducing immunosuppression regimens before such JCPyV variants infect the CNS. Other PML therapies that have proven effective, including VLP immunization or PD-1 blockade, could also be used to curtail the disease at the viremic stage [101,140,147,148].

## 11. Conclusions

Mutations in JCPyV VP1 likely occur at a low frequency during the infectious lifecycle, and increase with resurgent infection. The intimate evolution of PyVs with their natural hosts selects and retains VP1 sequences advantageous for viral transmission to uninfected hosts. For JCPyV, the archetype VP1 favors viral persistence in the urinary and GI tracts to facilitate egress of infectious virus via urine and feces, respectively. Yet, it is expected that the vast majority of mutations in VP1 are culled because they incur hits to viral replicative fitness, negate tropism for urinary and GI tracts, and are recognized by VP1 nAbs. In the setting of a weakened anti-viral T-cell response, PML-susceptible individuals may have a pre-existing or newly acquired VP1 nAb “blind spot,” such that discrete VP1 mutations could allow such viruses to outcompete archetype virus. The overlap of the virus’ receptor binding pocket with the targets of the nAb response causes many of these nAb-escape VP1 mutations to concomitantly alter or impair receptor binding and tissue tropism. Retention of brain tropism by some of these VP1 mutant viruses sets the stage for infection and injury in the CNS. Thus, a combination of altered receptor binding and antibody escape by JCPyVs carrying particular VP1 mutations may turn an asymptomatic kidney infection into a deadly brain disease.

## Figures and Tables

**Figure 1 viruses-12-01156-f001:**
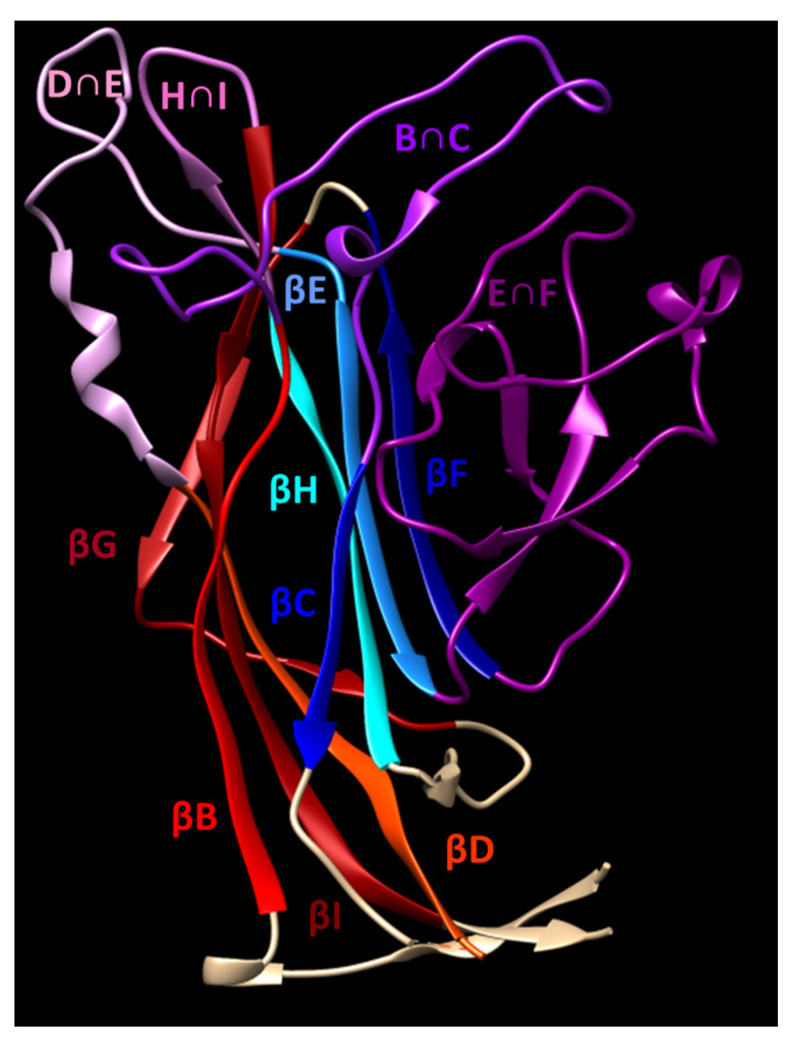
Solvent-exposed loops and β strands of JCPyV VP1. The exterior surface of VP1 is formed by the BC, DE, EF, and HI loops (shades of purple) connecting two sets of antiparallel β strands, BIDG and CHEF (shades of red and blue). Figure generated in UCSF Chimera using JCPyV VP1 structure 3NXG [18,22].

**Figure 2 viruses-12-01156-f002:**
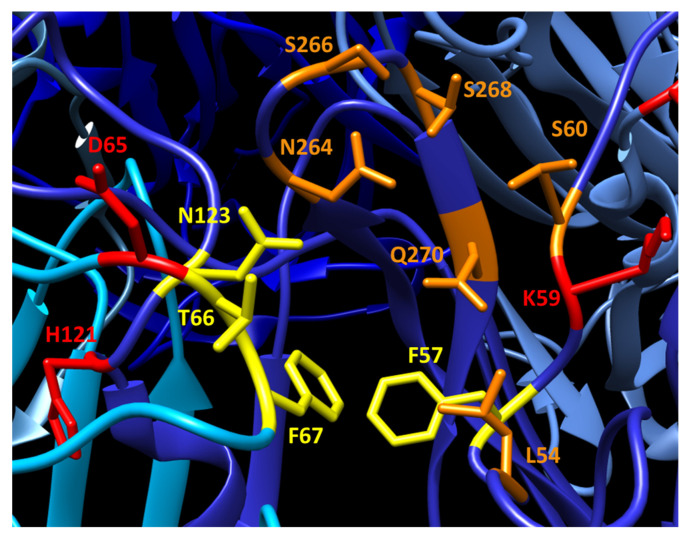
Overlap of receptor binding residues and locations of JCPyV-PML VP1 mutations. Side chains of VP1 amino acids that interact with LSTc are shown in yellow. Sites of JCPyV-PML VP1 mutations are shown in red. Residues that are both involved in LSTc binding and mutated in PML are shown in orange. LSTc interacting residues are assigned based on Neu et al. [18]. Indicated sites of PML mutations have been reported in several studies [76,84,85,101]. Neighboring VP1 subunits within the VP1 pentamer are denoted with shades of blue. Figure generated in UCSF Chimera using JCPyV VP1 structure 3NXG [18,22]. VP1 residue numbering throughout this article excludes the initial methionine.

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
