# Peer review of "JCPyV VP1 Mutations in Progressive Multifocal Leukoencephalopathy: Altering Tropism or Mediating Immune Evasion?"

_viruses, 2020, doi:10.3390/v12101156_

Round 1

Reviewer 1 Report

The review of M.D. Lauver and A.E. Lukacher "JCPyV VP1 Mutation in PML Altering or mediating tropism Immune Evasion?" Is very valuable summary of the current understanding of mutations in the major capsid protein VP1 and their importance for viral tropism, virulence and recognition by antibodies.

It is well organised and  written.

Comments:

I only disagree, that PyV virions leave ER as intact particles (page 3, lines74-75) as their conformation and arrangement of the minor structural proteins are significantly altered. The  paper cited as "23"   stated that   " cryo-electron tomography indicated that loss of inter-chain disulfides coupled with calcium depletion induces selective dissociation of the 12 vertex pentamers".

Author Response

The reviewer is absolutely correct.  We apologize that this error slipped through.  It has been corrected and the sentence revised as follows: "Capsid conformational changes and interactions with ER and cytosolic host proteins result in destabilization of VP1 pentamers and ejection of virions across the ER membrane into the cytosol." (lines 74-75).

Reviewer 2 Report

Revision manuscript “JCPyV VP1 Mutations in PML: Altering Tropism or Mediating Immune Evasion?” by Lauver MD & Lukacher AE

This is a comprehensive review of JCPyV VP1 mutations associated to the development of Progressive Multifocal Leukoencephalopathy. Their effects on viral tropism, immune evasion, and virulence are reviewed according to the most recent works on this topic. The review is of interest for scientists working in the field.

Author Response

We would like to thank the reviewer for his/her generous comments on our review.

Reviewer 3 Report

The manuscript by Lauver and Lukacher is a review article on the physiopathology of the human neurotropic Polyomavirus JCPyV entry into glial cells and the effect of the capsid protein VP1 mutations' in viral entry, propagation and immune evasion. It is a very well written paper, with a logical and sequential order, which is highly relevant to the field of virology and specifically Polyomaviruses. The literature cited is abundant and up to date, and the computer generated VP1 models depicted in the figures are elegant and very clear.

I only have a couple of minor comments and suggestions.

Page 1, Line 30: Although I fully agree that the most likely via of transmission is fecal/urine-oral, there is also evidence of a non-exclusive alternative route via the respiratory tract. Simple add that and a good reference supporting that fact (Monaco, J Virology, 1998)

Page 1, Line 36: Although JCPyV is classified by the IARC as a 2B agent (possibly carcinogenic to humans), is difficult to ignore the very large and compelling body of work showing its oncogenicity, both in brain tumors, as well, and better accepted in colon cancer, therefore I would either remove the last part of the sentence ("and is the only known oncogenic human polyomavirus"), or include a section describing both views. However, since that really would fall very much outside the scope of this manuscript, I believe is easier just o remove that last portion of the sentence.

Page 3, Line 104: The explanation of what the abbreviation EVs is (extracellular vessels) has not been described yet. Add "extracellular vessels" here and remove from later in Page 5, Line 195.

Page 4, Line 126: Astrocytes do not undergo lytic infection, oligodendrocytes do. Remove lytic, and perhaps add that astrocytes exhibit a transformed phenotype.

Page 4, Lines 127-128: The presence of apoptosis in PML is highly controversial. While only one study has detected it in human samples, and one in the chimeric mice, there are also papers that show non-apoptotic death (Seth, J Virology, 2004), and the activation of the anti-apoptotic protein Survivin in PML (Piña-Oviedo, AM J Pathology, 2010). This should also be discussed and these references added.

Again, these are just minor suggestions that would improve an excellent manuscript.

Author Response

We thank the reviewer for these complimentary comments on our review.

  1. We have added a sentence that PyVs may be transmitted via the respiratory tract and cited the indicated reference (line 32).
  2. We followed the reviewer's suggestion and deleted this part of the sentence.
  3. The spelled out abbreviation for EV is now at its first appearance in the text.
  4. We have replaced "lytic viral replication" with "a transformed phenotype" and added that astrocytes may undergo necrotic cell death after lytic infection Line 126).
  5. We have expanded on the the question of the mechanism of cell death by astrocytes and oligodendrocytes and included the cited references as well as a new reference showing that iPSC-derived human astrocytes support the complete JCPyV replication cycle (lines 127-130).